

# Association between visfatin and periodontitis: a systematic review and meta-analysis

Yaoqin Li[1],*, Caihong Xin[2],*, Jing Xie[3] and Xin Sun[3]

[1] Department of Stomatology, First Affiliated Hospital of Soochow University, Suzhou, China
[2] Department of Endocrinology and Metabolism, Fourth People's Hospital of Shenyang, Shenyang, China
[3] Department of Endocrinology and Metabolism, First Affiliated Hospital of Soochow University, Suzhou, China
* These authors contributed equally to this work.

## ABSTRACT

**Background:** Periodontitis is a chronic inflammatory disease caused by bacterial infection in the periodontal support tissue. Visfatin, a hormone secreted mainly by adipocytes and macrophages, plays an important role in immune regulation and defense. Although studies have indicated that patients with periodontitis have significantly high serum and gingival crevicular fluid levels of visfatin, the relationship between this adipocytokine and periodontal disease remains unclear.
**Aim:** The aim of this study was to systematically evaluate the association between visfatin levels and periodontitis.
**Methods:** The PubMed, Web of Science, ScienceDirect, EBSCO, and Wiley Online Library databases were searched for potential studies, using "periodontitis" and "visfatin" as the keywords in the title and abstract search fields. Standardized mean difference (SMD) values with corresponding 95% confidence intervals (CIs) were determined from the results of this meta-analysis.
**Results:** In total, 22 articles involving 456 patients with periodontitis and 394 healthy individuals (controls) were included in the meta-analysis. Visfatin levels were significantly higher in the patients with periodontitis than in the healthy individuals (SMD: 3.82, 95% CI [3.01–4.63]). Moreover, the visfatin levels were significantly lowered after periodontitis treatment (SMD: −2.29, 95% CI [−3.33 to −1.26]).
**Conclusion:** This first-ever meta-analysis comparing visfatin levels between patients with periodontitis and healthy individuals suggests that this adipocytokine can be a diagnostic and therapeutic biomarker for periodontal disease.

# INTRODUCTION

Periodontitis is a chronic inflammatory disease caused by bacterial infection of the periodontal support tissue. Bacteria and their toxic products damage periodontal tissue directly and trigger the host immune response, leading to alveolar bone absorption. Typical pathological features include the formation of periodontal pockets and alveolar

Corresponding author
Xin Sun, sunxin77@126.com

bone resorption, which ultimately lead to concerns such as tooth loss and chewing disorders (*Lim et al., 2020*). The incidence of periodontitis is approximately 10% worldwide and increases with age (*Janakiram & Dye, 2020*). In addition to the impact of the disease on the nutrition and quality of life of patients, its treatment places a significant burden on local economies, as the total worldwide economic cost of periodontitis in 2010 was reportedly 440 billion USD. Moreover, various systemic diseases have been shown to be correlated with periodontal disease (*Marcenes et al., 2013*). In 2012, the European Periodontal Disease Federation and the American Periodontal Disease Society jointly reported a relationship between periodontal and cardiovascular diseases. Additionally, studies have shown that periodontal disease may be a risk factor for premature delivery, stroke, Alzheimer's disease, diabetes, and other health conditions (*Sanz et al., 2020*).

Adipose tissue has traditionally been considered a long-term energy storage organ; however, researchers now realize its crucial role in metabolism. Generally, it is known to be an endocrine organ that secretes various bioactive substances/factors, collectively referred to as adipocytokines, which include visfatin, adiponectin, and leptin (*Unamuno et al., 2018*). Visfatin, a hormone secreted mainly by adipocytes and macrophages, plays a vital role in immune regulation and defense in humans. It stimulates the synthesis of inflammatory mediators and proteases in cells and inhibits the apoptosis of specific inflammatory cells. Studies have shown that patients with type 2 diabetes, obesity, metabolic syndrome, atherosclerosis, and cancer tend to have high plasma levels of visfatin (*Fukuhara et al., 2005*; *Jiang & Zhou, 2023*; *Yu et al., 2018*). Although it is well acknowledged that adiponectin and leptin levels are significantly high in patients with periodontal disease (*Zhu et al., 2017*), the relationship between visfatin and periodontitis remains unclear. Some studies have demonstrated that the serum and gingival crevicular fluid (GCF) levels of visfatin in patients with periodontitis are markedly higher than those in healthy individuals, whereas other reports have shown conflicting results (*Pradeep et al., 2011*; *Raghavendra et al., 2012*; *Xu et al., 2023*). Therefore, the objective of this study was to systematically evaluate the association between visfatin levels and periodontitis.

## METHODS

### Literature search

Potential studies published from 1980–2023 were searched for on the PubMed, Web of Science, ScienceDirect, EBSCO, and Wiley Online Library databases, with "periodontitis" and "visfatin" used as the keywords in the title and abstract search fields. The details of full search strategies were presented in File S1. The language limited to English. Additionally, the references of relevant literature were searched for publications that met the inclusion criteria and to eliminate duplicate studies. This meta-analysis was performed in accordance with the Preferred Reporting Items for Systematic Reviews and Meta-Analyses (PRISMA) guidelines and has been registered in PROSPERO under the accession number CRD42023455015. The PRISMA list is provided in Table S1.

**Table 1 Inclusion criteria/exclusion criteria of the selected studies in the meta-analysis.**

| Inclusion criteria | 1. Case-control or prospective cohort studies; |
|---|---|
| | 2. Studies reporting tde correlation between visfatin levels and periodontitis; |
| | 3. Studies written in English. |
| Exclusion criteria | 1. Reviews, case reports, comments, and experimental animal studies; |
| | 2. Duplicate or repeat publications; |
| | 3. Studies with insufficient data. |

## Inclusion criteria

Only studies that met the following criteria were included in this meta-analysis: (1) case-control or prospective cohort studies; (2) studies reported the correlation between visfatin levels and periodontitis; and (3) studies written in English (Table 1).

## Exclusion criteria

(1) Reviews, case reports, comments, and experimental animal studies; (2) duplicate or repeat publications; and (3) studies with insufficient data (Table 1).

## Data extraction and risk of bias

As part of the preliminary screening process, two reviewers (SX and LYQ) independently used the search strategy and read the titles and abstracts to exclude studies that did not meet the inclusion criteria. To determine whether the studies met the inclusion criteria, the two reviewers methodologically reviewed the full text. If the author's information is incomplete, they can contact and crosscheck the author. If the conclusions of the two evaluators were inconsistent, the differences were resolved through discussion. If the discussion failed to resolve any differences, it was judged and arbitrated by a third researcher (XCH). The diagnosis of periodontitis was established based on the clinical parameters of clinical attachment level (CAL), gingival index (GI), and probing pocket depth (PPD). Cochrane Collaboration recommends the Newcastle-Ottawa Scale (NOS) as a tool to assess bias in observational studies (*Higgins & Green, 2014*; *Wells et al., 2014*). Studies were rated according to the NOS, which ranged from zero to nine stars, and star scores were used to determine quality. There are three aspects in the NOS: the method for selecting case and control groups, their comparability, and the method for assessing exposure.

## Statistical analysis

We assessed heterogeneity among the included studies using the $I^2$ statistic and presented the data as standardized mean differences (SMDs) and 95% confidence intervals (CIs). Fixed-effects models were used if $I^2$ was <50% and heterogeneity among studies was low or moderate; otherwise, random-effects models were used if $I^2$ was >50%. A sensitivity analysis was performed to evaluate the stability of the results. Begg's and Egger's tests were used to detect publication bias. $P < 0.05$ was set as the significance level. Data analysis was performed using Stata version 12.0 (College Station, TX).

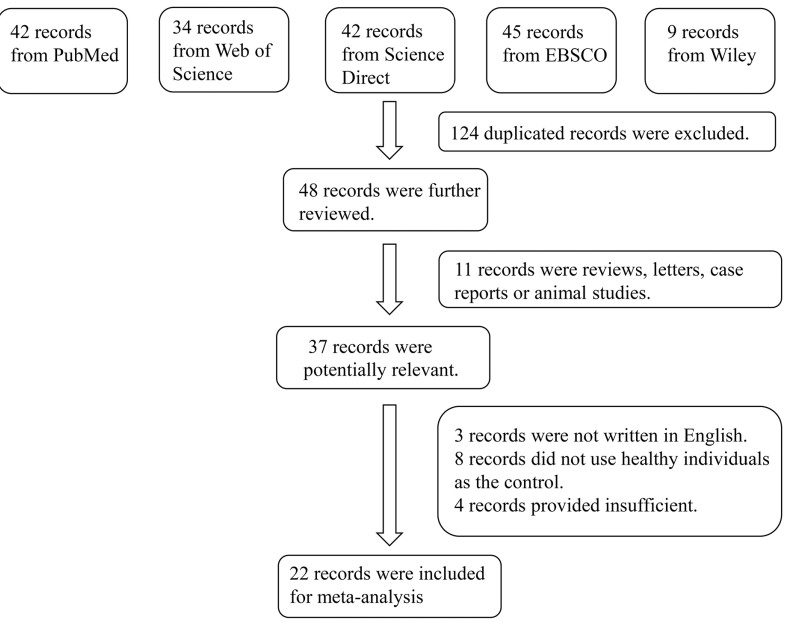

**Figure 1 Flowchart of the detailed procedure for the inclusion or exclusion of selected studies.**

# RESULTS

In total, 172 studies were retrieved from the PubMed, Web of Science, Embase, EBSCO, and Wiley Online Library databases. After screening, 124 duplicated studies were excluded. After reading the abstract of the remaining 48 studies, 37 of them were extracted as initially eligible studies. By examining full texts, 22 articles were finally included in the meta-analysis (*Pradeep et al., 2011*; *Raghavendra et al., 2012*; *Xu et al., 2023*; *Pradeep et al., 2012*; *Tabari et al., 2014*; *Abolfazli et al., 2015*; *Ghallab, Amr & Shaker, 2015*; *Mohamed et al., 2015*; *Tabari et al., 2015*; *Özcan et al., 2016a, 2016b, 2017*; *Tabari et al., 2018*; *Bahammam & Attia, 2018*; *Mopidevi et al., 2019*; *Rezaei et al., 2019*; *Çetiner et al., 2019*; *Paul et al., 2020*; *Saljoughi, Nasri & Bayani, 2020*; *Saseendran et al., 2021*; *Coutinho et al., 2021*; *Bengi et al., 2024*). The reasons for inclusion in the full-text selection are shown in Fig. 1. Overall, the 22 articles included 456 patients with periodontitis and 394 healthy individuals (controls). The characteristics of the selected studies are summarized in Table 2. According to the suggested criteria for the Selection, Comparability, and Exposure categories of the Newcastle-Ottawa Scale, the studies included in this meta-analysis were of acceptable quality (Table S2).

## Meta-analysis analysis

The meta-analysis results demonstrated that the visfatin levels in patients with periodontitis were significantly higher than those in healthy individuals (SMD:3.82, 95% CI [3.01–4.63]); the forest funnel plots are presented in Fig. 2. Visfatin levels were significantly lower in the patients with periodontitis after effective treatment (SMD: −2.29, 95% CI [−3.33 to −1.26]) (Fig. 3). In addition, these associations were highly heterogeneous.
**Table 2 The characteristic of the selected studies in the meta-analysis.**

| Author | Publication year | Study period | Region | Study design | Detection method | Case (n) | Control (n) | Sample | Details |
|---|---|---|---|---|---|---|---|---|---|
| Pradeep AR | 2011 | Mid-October 2009 to February 2010 | India | Case-control study | ELISA | 15 | 10 | GCF, serum | Case: CP patients, age range: 23–53 years; Control: patients with healthy periodontium; age range: 25–36 years |
| Pradeep AR | 2012 | November 2010 to February 2011 | India | Case-control study | ELISA | 10 | 10 | GCF, serum | Case: individuals with CP and without diabetes, 32.60 ± 7.59 years; Control: healthy individuals, 31.33 ± 3.48 years |
| Raghavendra NM | 2012 | Mid-September 2010 to January 2011 | India | Case-control and cohort study | ELISA | 15 | 15 | GCF, serum | Case: CP subjects who had signs of clinical inflammation 38.80 ± 5.63 years; Control: subjects with clinically healthy periodontium, 31.33 ± 3.48 years |
| Tabari ZA | 2014 | March 2012 to April 2013 | Iran | Case-control study | ELISA | 20 | 20 | Salivary | Case: CP patients, 6 males and 14 females, aged 32 to 62 years; Control: healthy individuals, 7 males and 13 females, aged 24 to 50 years |
| Mohamed HG | 2015 | July to December 2012 | Qatar | Case-control study | Multiplex biometric immunoassay | 44 | 30 | GCF | Case: CP patients, 18 males and 26 females, 55.37 ± 1.77 years; Control: healthy individuals, 15 males and 15 females, 47.15 ± 1.56 years |
| Ghallab NA | 2015 | September to December 2013 | Egypt | Case-control study | PCR | 20 | 10 | Tissue | Case: systemically healthy patients with CP, 48.3 ± 2.87 years; Control: systemically and periodontally healthy control group, 47.8 ± 2.9 years |
| Abolfazli N | 2015 | April 2012 to February 2013 | Iran | Cohort study | ELISA | 18 | 18 | Salivary, serum | patients with generalized moderate to severe CP, 41.16 ± 7.59 years |
| Tabari ZA | 2015 | March 2012 to June 2013 | Iran | Cohort study | ELISA | 20 | 20 | Salivary | patients with moderate to severe CP, 6 males and 14 females, 38.45 ± 9.98 years |
| Özcan E | 2016 | – | Turkey | Case-control study | ELISA | 27 | 18 | GCF | Case: CP patients, 19 males and 8 females, 41.59 ± 7.23 years; Control: healthy individuals, 12 males and 6 females, 37.72 ± 9.02 years |
| Özcan E | 2016 | 2013 to 2015 | Turkey | Case-control and cohort study | ELISA | 17 | 15 | Salivary | Case: patients with periodontitis, 9 males and 8 females, 41.52 ± 6.34 years; Control: healthy individuals, 6 males and 9 females, 44.06 ± 5.36 years |
| Özcan E | 2017 | – | Turkey | Case-control study | PCR | 21 | 19 | Tissue | Case: CP patients, 13 males and 8 females, 44.00 ± 8.25 years; Control: healthy individuals, 9 males and 10 females, 41.57 ± 14.4 years |

(Continued)

Li et al. (2024), *PeerJ*, DOI 10.7717/peerj.17187

| Author | Publication year | Study period | Region | Study design | Detection method | Case (n) | Control (n) | Sample | Details |
|---|---|---|---|---|---|---|---|---|---|
| Tabari ZA | 2018 | March 2014 to April 2015 | Iran | Case-control study | Immunohistochemistry | 20 | 20 | Tissue | Case: patients with moderate-to-severe CP, 14 females and 6 males, 42.58 ± 9.1 years; individuals with healthy periodontium, 12 females and 8 males, 33.5 ± 7.2 years |
| Bahammam MA | 2018 | – | Egypt | Case-control study | ELISA | 20 | 20 | GCF | Case: CP patients; Control: healthy individuals |
| Rezaei M | 2019 | August 2017 to September 2018 | Iran | Case-control study | ELISA | 15 | 15 | GCF | Case: CP patients, 45.03 ± 3.14 years; Control: healthy individual, 44.99 ± 3.26 years |
| Mopidevi A | 2019 | – | India | Case-control and cohort study | ELISA | 10 | 10 | Salivary | Case: moderate to severe CP, 43.90 ± 9.65 years; Control: periodontally healthy subjects with no underlying systemic diseases, 44.10 ± 8.41 years |
| ÇETİNER D | 2019 | October 2014 to January 2015 | Turkey | Case-control and cohort study | ELISA | 9 | 10 | GCF | Case: individuals who are normal weight generalized CP, 2 females and 7 males, 44.22 ± 6.74 years; Control: individuals of normal weight with no history of periodontitis, 8 females and 2 males, 27.80 ± 3.12 years |
| Saljoughi F | 2020 | August to November 2018 | Iran | Case-control study | ELISA | 23 | 30 | GCF | Case: participants without polycystic ovary syndrome but with advanced CP, all females, 45.3 ± 3.1 years; Control: healthy subjects, all females, 45.5 ± 3.3 years |
| Paul R | 2020 | – | India | Case-control study | ELISA | 30 | 30 | GCF | Case: patients with generalized CP; Control: patients who have healthy periodontium |
| Saseendran G | 2021 | February 2015 to July 2016 | India | Case-control and cohort study | ELISA | 16 | 16 | Salivary | Case: CP patients, 56.8 ± 8.0 years; Control: healthy individual, 19.2 ± 0.9 years |
| Coutinho A | 2021 | – | India | Case-control study | ELISA | 20 | 20 | Salivary | Case: systemically healthy subjects with generalized periodontitis, 35.90 ± 8.89 years; Control: systemically healthy subjects with healthy periodontium, 27.10 ± 3.92 years |
| Xu X | 2023 | March 2011 to March 2012 | China | Case-control study | ELISA | 40 | 20 | GCF, serum | Case: CP patients, 52.23 ± 9.65 years; Control: health individuals, 33.55 ± 6.98 years |
| Bengi VU | 2023 | – | Turkey | Case-control and cohort study | ELISA | 26 | 18 | GCF | Case: Stage III Grade B periodontitis patients, 17 males and 12 females, 42.9 ± 7.69 years; Control: periodontally healthy subjects, 12 males and 6 females, 37.7 ± 7.10 years |

**Note:**

ELISA, enzyme-linked immunosorbent assay; PCR, polymerase chain reaction; CP, chronic periodontitis; GCF, gingival crevicular fluid.

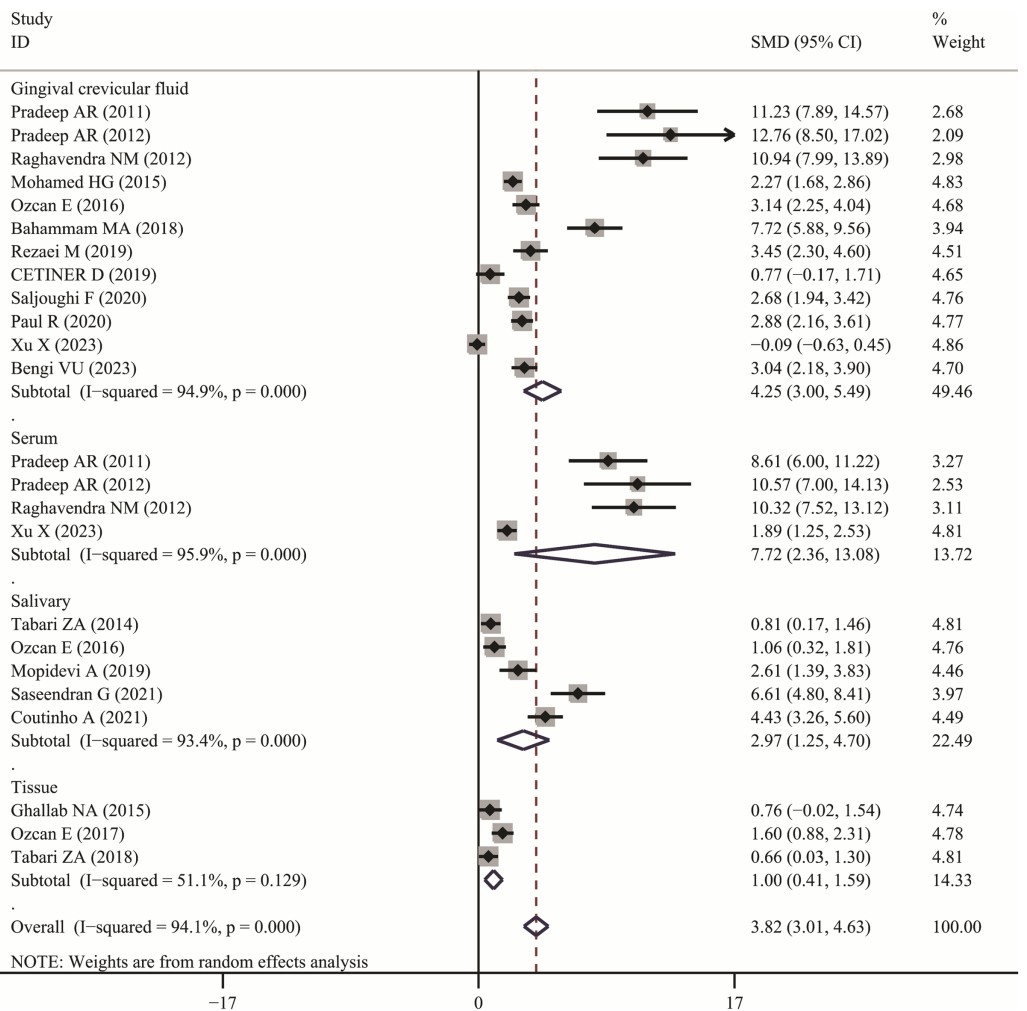

**Figure 2 Forest plots of visfatin levels in patients with periodontitis compared with healthy individuals.** Diamond represents the SMDs at 95% CI (*Pradeep et al., 2011*, *2012*; *Tabari et al., 2018*; *Raghavendra et al., 2012*; *Özcan et al., 2016a*, *2016b*; *Bahammam & Attia, 2018*; *Rezaei et al., 2019*; *Çetiner et al., 2019*; *Saljoughi, Nasri & Bayani, 2020*; *Paul et al., 2020*; *Xu et al., 2023*; *Bengi et al., 2024*; *Mopidevi et al., 2019*; *Coutinho et al., 2021*; *Ghallab, Amr & Shaker, 2015*; *Özcan et al., 2017*; *Mohamed et al., 2015*; *Tabari et al., 2014*; *Saseendran et al., 2021*).

In subgroup analysis, we confirmed the visfatin levels in patients with periodontitis were significantly higher of different sample (GCF, SMD: 4.25, 95% CI [3.00–5.49]; Serum, SMD: 7.72, 95% CI [2.36–13.08]; Salivary, SMD: 2.97, 95% CI [1.25–4.70]; Tissue, SMD: 1.00, 95% CI [0.41–1.59]). Only two articles used serum visfatin as the therapeutic biomarker for periodontitis. In other studies, compared with before treatment, visfatin levels in saliva and GCF of patients with periodontitis significantly decreased (GCF, SMD: −3.04, 95% CI [−5.98 to −0.10]; Salivary, SMD: −1.64, 95% CI [−2.75 to −0.53]).

## Sensitivity analysis and publication bias

Each study was subjected to a sensitivity analysis to determine its influence. Sensitivity analysis showed no significant differences from our previous estimates, indicating that a

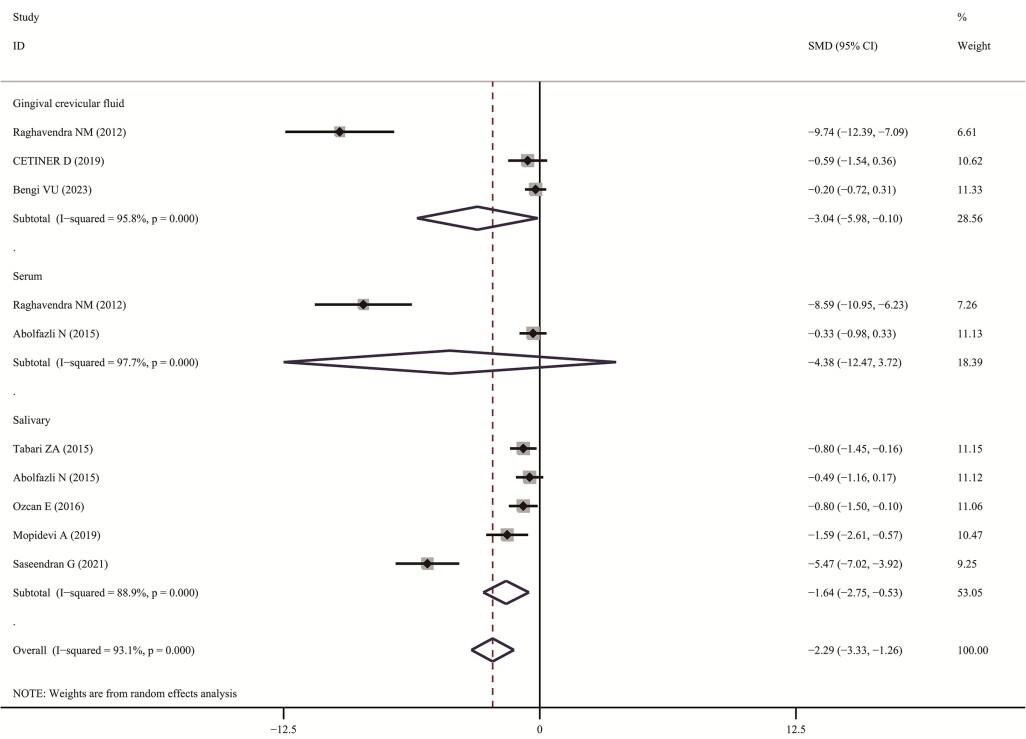

**Figure 3 Forest plots of visfatin levels in patients with periodontitis before and after treatment** (*Raghavendra et al., 2012*; *Çetiner et al., 2019*; *Bengi et al., 2024*; *Abolfazli et al., 2015*; *Tabari et al., 2015*; *Özcan et al., 2016a*, *2017*; *Mopidevi et al., 2019*; *Saseendran et al., 2021*).

single study had a marginal impact on the overall estimate (Figs. 4 and 5). Accordingly, the meta-analysis yields stable results. A thorough and comprehensive search of the databases was conducted. Begg's and Egger's tests were conducted to identify whether publication bias was present in the reviewed studies. The results ($P > 0.05$) indicated that there was no publication bias.

## DISCUSSION

This is the first-ever meta-analysis to compare visfatin levels between patients with periodontitis and healthy individuals. Although some studies have examined the visfatin levels in patients with periodontitis, the inconsistent results obtained have made establishing the relationship between the adipocytokine and disease challenging. Eighteen independent studies were included in our present meta-analysis, with the results indicating that visfatin levels in patients with periodontitis were significantly higher than those in healthy individuals and substantially lowered after disease treatment.

Visfatin was first isolated from a human peripheral blood lymphocyte cDNA library in 1994. It is known as a pre-B-cell clone enhancer (PBEF) owing to its synergistic effect on stem cell factor- and interleukin (IL)-7-mediated B-cell maturation (*Samal et al., 1994*). In 2002, *Rongvaux et al. (2002)* discovered that the mouse homolog of the human *PBEF* gene encodes nicotinamide phosphoribosyltransferase (NAMPT), which catalyzes the condensation of nicotinamide with 5-phosphate ribose-1 pyrophosphate, producing

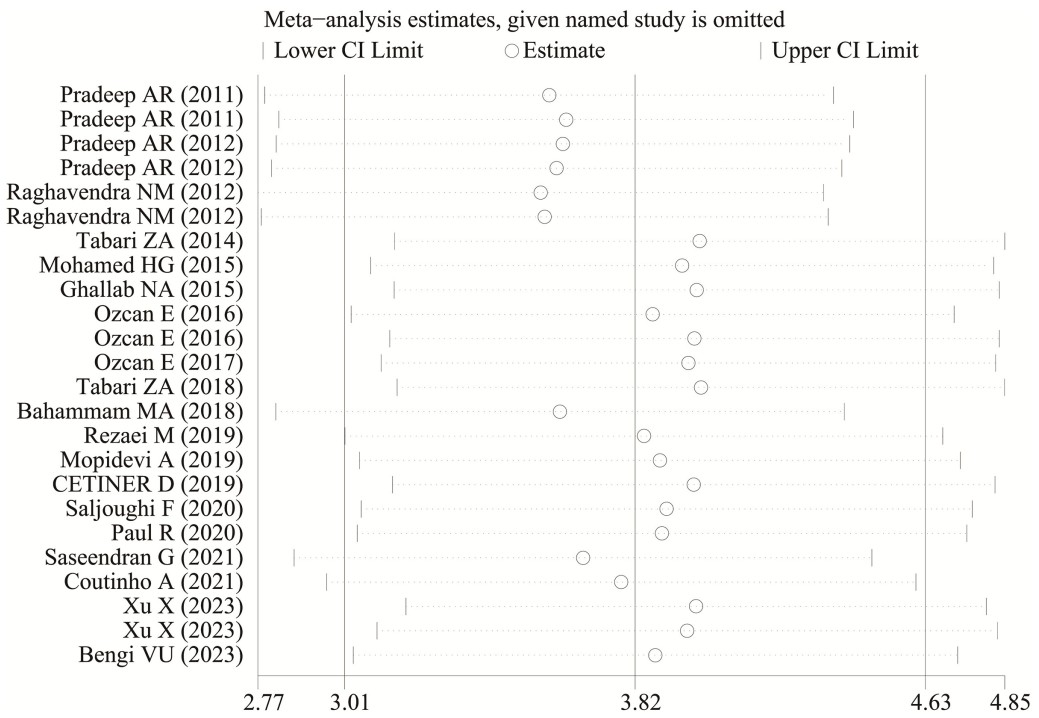

**Figure 4 The sensitivity analysis results of visfatin levels in patients with periodontitis compared with healthy individuals** (*Pradeep et al., 2011, 2012; Raghavendra et al., 2012; Tabari et al., 2014; Mohamed et al., 2015; Ghallab, Amr & Shaker, 2015; Özcan et al., 2016a, 2016b, 2017; Tabari et al., 2018; Bahammam & Attia, 2018; Rezaei et al., 2019; Mopidevi et al., 2019; Çetiner et al., 2019; Saljoughi, Nasri & Bayani, 2020; Paul et al., 2020; Saseendran et al., 2021; Coutinho et al., 2021; Xu et al., 2023; Bengi et al., 2024*).

nicotinamide mononucleotide (hence the protein symbol NAMPT), an intermediate product in nicotinamide adenine dinucleotide biosynthesis. The visfatin-encoding gene, which is located on chromosomes 7q22.1–7q31.33, is 34.7 kb long and contains 11 exons and 10 introns. Visfatin is highly expressed in the bone marrow, liver tissue, muscle, brown adipose tissue, liver, and kidney, whereas it is less expressed in the white adipose tissue, lungs, spleen, testes, and muscle tissue (*Fukuhara et al., 2005*); its expression has been also observed in other tissues (*Revollo et al., 2007*). Its widespread expression reflects its key biological role.

The fact that patients with periodontitis have higher levels of visfatin is likely related to periodontal pathogens promoting its synthesisc in the periodontal tissue. *Özcan et al. (2016b)* collected GCF and plaque samples from *Porphyromonas gingivalis*-infected patients with periodontitis and healthy individuals and found that the GCF concentration of visfatin was significantly higher in the individuals infected with the bacterium. A significant positive correlation was observed between visfatin concentration and *P. gingivalis* infection. In another study, visfatin expression was significantly increased in periodontal ligament cells that had been exposed to *Fusobacterium nucleatum*, whereas it was inhibited in cells that had been preincubated with antibodies to neutralize Toll-like receptors (TLR) two and four (*Nogueira et al., 1994*). Additionally, gingival fibroblasts stimulated with *P. gingivalis* or *F. nucleatum* showed significantly increased visfatin

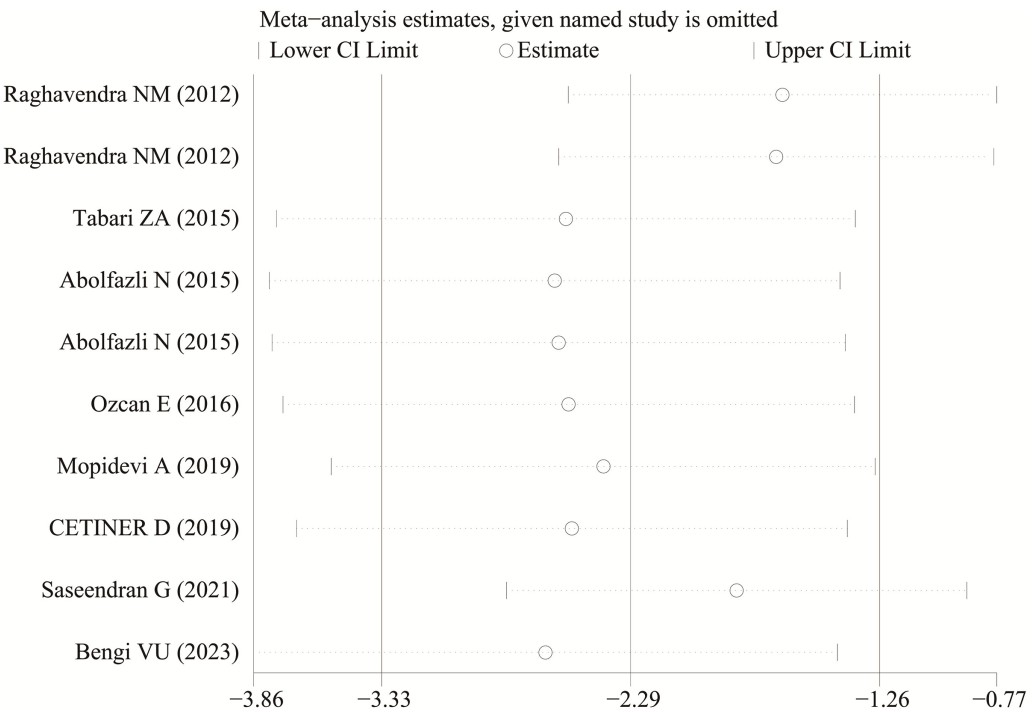

Meta−analysis estimates, given named study is omitted

**Figure 5 The sensitivity analysis results of visfatin levels in patients with periodontitis before and after treatment** (*Raghavendra et al., 2012*; *Tabari et al., 2015*; *Abolfazli et al., 2015*; *Özcan et al., 2016b*; *Mopidevi et al., 2019*; *Çetiner et al., 2019*; *Saseendran et al., 2021*; *Bengi et al., 2024*).

expression, whereas the effect disappeared after inhibition of the mitogen-activated protein kinase (MAPK) and nuclear factor-kappa B (NF-κB) signaling pathways (*Damanaki et al., 2014*). These results suggest that periodontal pathogens and their metabolic products can upregulate visfatin expression by binding to TLR, activating the MAPK and NF-κB signaling pathways.

Visfatin is used as an inflammation biomarker in many diseases (*Mermutluoglu & Tekin, 2023*; *El-Masry et al., 2024*). Inflammatory cytokines, including IL-1β, IL-6, tumor necrosis factor-alpha, and chemokine (C-C motif) ligand 2 (CCL2), play a crucial role in the inflammatory response of periodontitis (*Yucel-Lindberg & Bage, 2013*). Matrix metalloproteinases (MMPs) have been shown to be closely related to the destruction of periodontal tissue and participate in the degradation of the extracellular matrix in the disease (*Xiang et al., 2009*). Visfatin upregulates *MMP-1* and *CCL2* expression in periodontal ligament cells. Additionally, the upregulation of *MMP-1* and *CCL2* gene expression by endorphins increased their respective protein levels, indicating consistency between the two (*Nokhbehsaim et al., 2013*). MMP-1 alters the cellular microenvironment by degrading collagen fibers in the extracellular matrix. After nonsurgical periodontal treatment, a strong correlation exists between decreases in the MMP and visfatin levels (*Özcan et al., 2016a*). These results suggest that visfatin promotes the expression of MMPs in periodontal cells, exacerbating damage to the periodontal tissue. CCL2, a chemotactic protein produced by different cell types, plays a vital role in regulating the migration and

infiltration of monocytes, T lymphocytes, and natural killer cells. Additionally, CCL2 is involved in T-cell immunity, and its high expression can be detected at sites of periodontitis. Therefore, visfatin may exacerbate periodontal tissue damage by promoting the production of these molecules. Moreover, patients with systemic diseases, such as obesity, may experience adverse effects on the periodontal tissue due to the high expression of adiponectin in their bodies (*Nokhbehsaim et al., 2014*). As mentioned previously, visfatin upregulates the expression of CCL2 in periodontal ligament cells, thereby exacerbating local inflammatory responses. However, the exact mechanism by which visfatin promotes the upregulation of inflammatory cytokines remains unclear. One study on endothelial cell dysfunction found that visfatin can activate the NF-κB signaling pathway in the cells to promote the expression of inflammatory cytokines (*e.g.*, IL-6 and IL-8) (*Nokhbehsaim et al., 2013*). Further research is needed to confirm the existence of this mechanism in periodontal tissue.

In a previous study, the regenerative treatment environment was stimulated by applying enamel matrix proteins to periodontal ligament cells in the presence and absence of visfatin. The results showed that the enamel matrix proteins promoted the synthesis of vascular endothelial growth factor and transforming growth factor (TGF) while inducing the secretion of matrix molecules such as collagen and periosteal protein, thereby promoting healing (*Lee et al., 2009*). Collagen and periosteal proteins are involved in periodontal tissue healing and are highly expressed in periodontal ligament cells. They play essential roles in regulating cell proliferation, differentiation, and adhesion. Enamel matrix proteins include bone morphogenetic proteins (BMPs) and TGF, which promote periodontal tissue regeneration. Many cell surfaces contain BMP and TGF receptors. Visfatin may regulate the expression of these receptors in periodontal ligament cells, thereby affecting periodontal tissue healing. Visfatin and enamel matrix proteins regulate the expression of BMP and TGF receptors in opposing manners, with the former inhibiting their expression and the latter upregulating them. Overall, enamel matrix proteins in periodontal ligament cells promote periodontal tissue regeneration, whereas visfatin inhibits enamel matrix protein activity.

Our study had some limitations. Most of the studies included in this meta-analysis had small sample sizes, rendering the total sample size insufficient for a more definitive conclusion of the findings. Furthermore, the severity of periodontitis varied among the different studies, and different treatment durations and methods were used. These factors may have affected the meta-analysis results. Therefore, the results presented herein should be interpreted with caution, as further research is required to verify them.

## CONCLUSION

To the best of our knowledge, this meta-analysis is the first to compare visfatin levels between patients with periodontitis and healthy individuals. Our findings suggest that the levels of this adipocytokine are substantially altered in patients with periodontal disease. Hence, visfatin is a potential inflammatory factor that acts as a mediator in the pathogenesis of periodontitis and may serve as a diagnostic and therapeutic biomarker for this oral disease.

## ACKNOWLEDGEMENTS

We thank Editage for English-language editing.

### Funding

The authors received no funding for this work.

### Competing Interests

The authors declare that they have no competing interests.

### Author Contributions

- Yaoqin Li conceived and designed the experiments, analyzed the data, prepared figures and/or tables, and approved the final draft.
- Caihong Xin performed the experiments, analyzed the data, authored or reviewed drafts of the article, and approved the final draft.
- Jing Xie performed the experiments, prepared figures and/or tables, and approved the final draft.
- Xin Sun analyzed the data, authored or reviewed drafts of the article, and approved the final draft.

### Data Availability

This is a systematic review and meta-analysis.

### Supplemental Information

Supplemental information for this article can be found online at http://dx.doi.org/10.7717/peerj.17187#supplemental-information.

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
