# Peer review of "Association between visfatin and periodontitis: a systematic review and meta-analysis"

_PeerJ, doi:10.7717/peerj.17187_

## Round 0.1 · original submission · Minor Revisions

1. There is no such database called Willy database. Could you please further clarify this?

2. Present the IC/EC in a tabular format

3. There is no table for critical appraisal of the selected studies.

4. Was there any timeline criteria to select the studies?

**Language Note:** The review process has identified that the English language must be improved. PeerJ can provide language editing services - please contact us at copyediting@peerj.com for pricing (be sure to provide your manuscript number and title). Alternatively, you should make your own arrangements to improve the language quality and provide details in your response letter. – PeerJ Staff

·

Basic reporting

Revision is necessary to improve English language writing.

The introduction part needs to be rewritten with more relevant articles.

Experimental design

The search strategy must be renewed as this study did not cover all relevant articles on visfatin and periodontitis.

This study did not include any new relevant articles that were published at the end of 2023.

Validity of the findings

There needs to be more discussion on relevant articles to complete the short discussion part.

Additional comments

The connection between visfatin and periodontitis is a hot topic in this new and hot topic manuscript, but there are some methodological shortcomings that were addressed previously.

·

Basic reporting

There are a lot of grammatical and punctuation mistakes in both the text and the figures. I would recommend the article to be proofread by someone adept in scientific writing. Some parts are very confusing owing to the language used and need to be clarified.

Sufficient background has been provided and the review appears to have been conducted satisfactorily.

Experimental design

No comment.

Validity of the findings

No comment.

Additional comments

In this article, the authors performed a systematic review to systematically evaluate the association between visfatin level and periodontitis. While the idea is good and the review has been conducted satisfactorily, I have some thoughts on how this manuscript could be improved and made fit for publication.

1) The specific key words and combinations used should be stated clearly.
2) Lines 31-32 state that “After screening, 22 articles comprising 456 patients with periodontitis and 394 healthy individuals were included in the meta-analysis”. Can this line be elaborated?
3) The conclusion of the abstract (lines 35-37) states “This is the first meta-analysis to compare visfatin levels between patients with periodontitis and 36 healthy individuals. Thus, visfatin levels can be considered diagnostic and therapeutic biomarkers for periodontal diseases.” The second part can be phrased better. For example, you could say, “This review found that visfatin levels may be considered……”
4) Line 87-88 needs to be modified. The tense of the text needs to be consistent.
5) The language used in Figure 1 is unclear and needs to be fixed.

Reviewer 3 ·

Basic reporting

clear and professional

Experimental design

Systematic search strategy including focus question, strategy Population , exposure, Search combination, Database search Electronic, journals not included
Assessment of case definition and data synthesis not mentioned

Validity of the findings

Systematic search strategy including focus question, strategy Population , exposure, Search combination, Database search Electronic, journals not included
Assessment of case definition and data synthesis not mentioned

conclusion linked with the aim of the study

Additional comments

Minor Revision needed

---

## Round 0.2 · accepted · Accept

Thanks for making the changes.

·

Basic reporting

The language writing was revised and many of typographical and punctuation mistake was corrected.

Experimental design

All relevant articles are retrieved.

Validity of the findings

Discussion part was rewrote and completed.

Reviewer 3 ·

Basic reporting

clear and unambiguous

Experimental design

research question needs to be defineds

Validity of the findings

Search strategy is proper